# Effect of the Combination of Probiotics and Korean Red Ginseng on Diabetic Wound Healing Exposed to Diesel Exhaust Particles(DEPs)

**DOI:** 10.3390/medicina59061155

**Published:** 2023-06-15

**Authors:** Hye Min An, Young Suk Choi, Sung Kyoung Bae, Young Koo Lee

**Affiliations:** 1Department of Medical Sciences, Soonchunhyang University, Asan-si 31538, Republic of Korea; ahm112597@naver.com (H.M.A.); qotjdrud00@naver.com (S.K.B.); 2Department of Orthopedic Surgery, Soonchunhyang University Bucheon Hospital, Bucheon-si 14584, Republic of Korea; 3Department of Biology, Soonchunhyang University, Asan-si 31538, Republic of Korea

**Keywords:** diabetic wound, diesel exhaust particles (DEPs), wound healing, probiotics, Korean red ginseng, tissue engineering

## Abstract

*Background and Objectives:* Diesel exhaust particles (DEPs) are a major component of air pollution and adversely affect respiratory and cardiovascular disease and diabetic foot ulcers if diabetic patients are exposed to them. There are currently no studies on treating diabetic wounds exposed to DEPs. So, the effect of a combination of probiotics and Korean red ginseng on a diabetic wound model exposed to DEPs was confirmed. *Materials and Methods:* Rats were randomly divided into three groups according to DEP inhalation concentration and whether they underwent applications of probiotics (PB) and Korean red ginseng (KRG). Wound tissue was collected from all rats, and wound healing was evaluated using molecular biology and histology methods. *Results:* The wound size of all groups decreased over time, but there was no significant difference. As a result of the molecular biology experiment, the expression of NF-κB p65 on day 7 was significantly higher in group 2 than in the normal control group. As a result of histological analysis, unlike the primary control group, it was confirmed that granule tissue was formed on the 14th day in the normal control group and group 2. *Conclusions:* The findings in this study suggest that combined treatment with PB and KRG can promote the healing of DEP-exposed diabetic wounds.

## 1. Introduction

In recent decades, the prevalence of diabetes has considerably increased because of rapid urbanization and a sedentary lifestyle. Approximately 693 million people are expected to become diabetic by 2045; 49.7% of these people will not receive a formal diagnosis of diabetes [1]. Diabetes, a chronic disease characterized by high blood sugar levels, is divided into two types: 1 (insulin deficiency) and 2 (insulin resistance). Diabetes can cause various complications over time, including neuropathy, retinopathy, nephropathy, and diabetic foot disease. Among patients with diabetes worldwide, the prevalence of diabetic foot ulcers is 6.3%; amputation is required in 0.03–1.5% of cases [2,3]. Diabetic foot ulcers, which occur 10-fold to 30-fold more frequently in patients with diabetes, are the most serious and costly complication of diabetes [4].

Factors such as smoking and drinking can worsen diabetic ulcers, and, recently, DEPs have been known to adversely affect diabetes. DEPs, generated by factory smoke and automobile exhaust gases, are divided into three types according to particle size: PM 10.0 (≤10 μm; coarse dust), PM 2.5 (≤2.5 μm; fine dust), and PM 0.1 (≤0.1 μm; ultrafine dust) [5]. DEPs enter the body through the respiratory tract and are transmitted to the circulatory system through blood circulation, oxidative stress increases through the expression of ROS due to mitochondrial damage, and inflammatory reactions occur through the expression of pro-inflammatory cytokines [6,7].

Wound healing consists of hemostasis, inflammation, proliferation, and remodeling; in chronic injuries caused by diabetes or pathogen infection, the transition from inflammation to proliferation is unsuccessful, and healing is delayed [8]. *Staphylococcus aureus*, a major cause of infection, is commonly found in both normal and chronic wounds, where it delays healing [9,10].

In general, probiotics and Korean red ginseng (KRG) are well known to promote the healing of acute wounds.

According to the World Health Organization and the Food and Agriculture Organization of the United Nations, probiotics are defined as “living microorganisms that give health benefits to hosts when administered in appropriate amounts” [11]. Oral administration of probiotics regulates the host immune response to achieve intestinal harmony, strengthen barriers, and alleviate allergic and inflammatory skin diseases [12]. Probiotic bacteria kill pathogens, strengthen epithelial barriers, and promote cytokine production through skin wound healing, epithelial cell migration and function, and blast induction [13]. Representative probiotics include lactic acid bacteria, bifidobacterium, and Lactobacillus. Furthermore, *Lactobacillus rhamnosus* GG (LGG) reportedly heals ulcers by stimulating angiogenesis and inducing proteoglycan deposition [14].

For thousands of years, KRG has been used as a nutritional and therapeutic agent for various diseases in Asian countries such as Korea, Japan, and China. KRG has positive effects such as immune enhancement, vitality recovery, fatigue relief, blood flow improvement, antioxidant properties, and memory improvement. KRG contains ginsenosides and saponins such as Rg1, Rb1, and Rb2, which play important roles in antioxidant activity [15]. Additionally, saponins and ginsenosides activate anti-inflammatory responses that can reduce edema and skin inflammation, stimulate the formation of new blood vessels, and activate the production of vascular endothelial growth factor and the pro-inflammatory cytokine interleukin (IL)-1β [16]. KRG promotes wound healing by increasing the expression of transforming growth factor-β (TGF-β) and vascular endothelial growth factor (VEGF) in the early stage of wound healing, as well as the expression of matrix metalloproteinase (MMP)-1 and MMP-9 in the late stage of wound healing [17].

The use of probiotics and KRG in the treatment of acute wounds promotes healing by decreasing the expression level of inflammatory cytokines (e.g., IL-1β and TNF-α) [18]. In addition, oral administration of fermented probiotics and KRG to rat diabetes wound models improved typical diabetes symptoms (e.g., reduced blood sugar, increased weight, and increased glycated hemoglobin (HbA1c)). Thus, the combined treatment is reportedly useful and effective for the treatment of diabetes.

Many studies have been carried out to promote healing of diabetic wounds, but few have been carried out to promote healing of diabetic wounds exposed to DEPs.

Since there is evidence that applying probiotics and red ginseng to general and chronic wounds, respectively, accelerates wound healing, we hypothesize that the combination of probiotics and red ginseng will promote wound healing by controlling the expression of cytokines and growth factors in diabetic wounds exposed to fine dust. Therefore, in this study, the effect of probiotics and red ginseng on diabetic wounds in fine-dust-exposure environments will be evaluated through the expression analysis of tumor necrosis factor (TNF)-α, nuclear factor (NF)-κB, MMP-9, and TIMP-1.

## 2. Materials and Methods

### 2.1. Animal and Induction of Diabetes

Sprague–Dawley (SD; 36 male) rats aged 6 weeks (weight, 155–200 g) were purchased from Nara-Bio (Seoul, Korea). Rats were housed in cages with a 12 h/12 h light/dark cycle at a temperature of 23 ± 2 °C and humidity of 50 ± 20% for 2 weeks before the experiment; they were given free access to food and water. Diabetes was induced in SD rats by intraperitoneal injection of 60 mg/kg streptozotocin (STZ) in 0.1 M sodium citrate after 8 h of fasting. At 24 h after STZ injection, blood was collected from the tail vein and blood glucose was measured using the Accu-Chek^®^ System (Roche Diagnostics, Mannheim, Germany). Rats were diagnosed with diabetes when they exhibited a blood glucose level of >300 mg/dL at Figure 1.

### 2.2. Methicillin-Resistant S. aureus (MRSA) Infection Wound Induction and DEP Inhalation

SD rats received intraperitoneal injections of a 4:1 mixture of zolazepam/tiletamine and xylazine. After each rat’s abdominal hair had been shaved and the surgical site had been sterilized with 70% ethanol, sterilizing scissors were used to induce a 2.5 cm thick skin defect. Next, a donut-shaped splint was created using a 1 mm thick polypropylene (pp) material with a diameter of 5 cm. To prevent contraction between the wound edge and healing tissue, a splint was placed in the center of the wound and fixed using nylon sutures (Prolene; W8434; Ethicon, Inc., Somerville, NJ, USA). Standard strains of MRSA were incubated overnight at 37 °C on MRSA screen agar. MRSA was diluted with sterile saline in the McFarland 0.5 standard turbidity meter (HI 93703; Hanna Instruments Inc., Woonsocket, WI, USA) and applied to wound. To prevent the wound from drying out, Vaseline (Samhyun Pharmaceutical, Seoul, Korea) was applied to the wound using gauze; this was followed by the application of Opsite-film (Opsite Flexifix; Smith & Nephew Medical Ltd., Hull, UK) and a bandage. The Vaseline-covered gauze was replaced at 3-day intervals throughout the 14-day experimental period. Rats were exposed to DEPs in a 350 mm hemispherical exposure chamber (inner diameter, 342 mm; outer diameter, 383 mm; height, 200 mm) sealed with an acrylic material for 1 h per day over a period of 13 days, beginning the day after wound formation as shown in Figure 2 and Figure 3.

### 2.3. Treatment with a Combination of Probiotics and KRG

SD rats were randomly divided into the following groups: normal control, 0 µg/m^3^ DEP inhalation (*n* = 12); group 1, 80 µg/m^3^ DEP inhalation (*n* = 12); and group 2, 80 µg/m^3^ DEP inhalation + administration of *L. rhamnosus* GG and KRG powder (*n* = 12). Once per day, *L. rhamnosus* GG at a concentration of 1 × 10^9^ colony-forming units/mL and KRG powder at a concentration of 200 mg/kg were diluted in 1 mL Dulbecco’s phosphate-buffered saline, then administered orally. During the 13-day trial period, dressing changes were performed at 3-day intervals. Experimental design is shown in Figure 4.

### 2.4. Skin Biopsy

At 4, 7, 10, and 14 days after wounding, rats were euthanized and skin biopsy specimens were acquired. Skin tissue was collected up to the edge of the normal skin surrounding the wound. A portion of the tissue was fixed in 4% paraformaldehyde, then processed into paraffin blocks for histology experiments. The remaining portion of tissue was immediately frozen in liquid nitrogen and stored at −80 °C for molecular biology analysis.

### 2.5. Wound Size Measurement

On days 1, 4, 7, 10, and 14 after wound induction, a sterilized transparent film was attached to the wound, and the wound was measured along the edge. The wound size was measured using ImageJ software (v1.53t, National Institutes of Health, Bethesda, MD, USA). A square shape with a size of 1 cm^2^ was scanned and recorded as a reference, and the wound area was converted into a percentage.

### 2.6. Western Blotting

Harvested skin tissue from each time point was analyzed using complete ethylenediaminetetraacetic acid-free protease inhibitor cocktail (11836170001; Roche Diagnostics GmbH), PhosSTOP phosphatase inhibitor (30498800; Sigma-Aldrich, St. Louis, MO, USA), and radioimmunoprecipitation assay buffer (R4200; GenDEPOT, Barker, TX, USA) to separate the protein. Protein concentrations were determined using the Lowry method and then quantified as 100 μg. Proteins were separated by 10% sodium dodecyl sulfate-polyacrylamide gel electrophoresis, then transferred to polyvinylidene difluoride membranes (10600023; GE Healthcare Life Sciences, Chalfont, UK). After membranes had been blocked with 5% skim milk (232100; Becton Dickinson, Franklin Lakes, NJ, USA) for 1 h at room temperature, they were incubated with anti-NF-κB p65 polyclonal antibody (ab16502; 1:300 dilution; Abcam; Cambridge, UK), anti-TNF-α monoclonal antibody (SC-52746; 1:300 dilution; Santa Cruz Biotechnology, Santa Cruz, CA, USA), anti-MMP-9 polyclonal antibody (ab76003, 1:500 dilution, Abcam), anti-TIMP-1 monoclonal antibody (SC-21734; 1:1000 dilution; Santa Cruz Biotechnology), or anti-β-actin polyclonal antibody (4970S; 1:5000 dilution; Cell Signaling; Beverly, MA, USA) for 24 h at 4 °C. Anti-data are shown in Table 1. Next, membranes were incubated with the appropriate secondary antibody (SC-2005 (1:3000 dilution) or SC-2030 (1:5000 dilution); Santa Cruz Biotechnology) for 1 h at room temperature. Bands were visualized by enhanced chemiluminescence (ECL 2232; GE Healthcare Life Sciences; Chalfont, UK) and a C280 bioimaging system (Azure Biosystems; Dublin, CA, USA). Band intensities were normalized to β-actin and assessed using ImageJ 1.53 software (National Institutes of Health; Bethesda, MD, USA).

### 2.7. Histology Analysis of Hematoxylin and Eosin (H&E) Staining

At 4, 7, 10, and 14 days after wound induction, tissue was collected, paraffinized, cut into 4 μm sections, placed on coated slides, stained with hematoxylin and eosin (H&E; Gills hematoxylin, 1% eosin; Muto, Tokyo, Japan), and imaged at 100× and 200× magnification using a light microscope (DXM1200; Nikon, Tokyo, Japan). Granulation tissue formation and inflammatory cell infiltration were scored according to Table 2 and Table 3.

### 2.8. Histology Analysis of Immunohistochemistry (IHC) Staining

At each time point, tissue was collected, paraffinized, cut into 4 μm sections, placed on coated slides, and incubated at 60–70 °C for 30 min. Tissue slides were deparaffinized in xylene and ethanol (100%, 95%, 80%, and 70%), then washed with 1× Tris-buffered saline and 1.5% normal serum (PK-6102 or PK-6101; Vector Laboratories, Burlingame, CA, USA) for 1 h. Next, they were incubated overnight at 4°C with anti-NF-κB p65 polyclonal antibody (ab16502; 1:100 dilution; Abcam; Cambridge, UK), anti-TNF-α monoclonal antibody (SC-52746; 1:100 dilution; Santa Cruz Biotechnology; Santa Cruz, CA), anti-MMP-9 polyclonal antibody (ab76003; 1:100 dilution; Abcam), or anti-TIMP-1 monoclonal antibody (SC-21734; 1:100 dilution; Santa Cruz Biotechnology) in blocking buffer. Tissue slides were then washed with 1× Tris-buffered saline and incubated with biotinylated secondary antibody (PK-6102 or PK-6101; 1:200 dilution; Vector Laboratories; Burlingame, CA) for 1 h at room temperature. Subsequently, they were incubated for 30 min with Vectastain Elite Avidin-Biotin Complex (PK-6102 or PK-6101; Vectastain^®^; Vector Laboratories) to increase the target antigen–antibody response. Finally, tissue sections were stained with chromogen diaminobenzidine (C02-100; Liquid DAB+ Chromogen Kit; Golden Bridge International Inc., Mukilteo, WA, USA) for 10-15 min and with hematoxylin for 3-5 min. After staining, the tissue slides were dried and photographed at 100× and 200× magnification using a light microscope. Two pathologists scored the expression intensity of each cytokine and the whiteness of stained cells, as shown in Table 4.

### 2.9. Statistical Analysis

The Mann–Whitney U test was used for comparisons involving ≥3 samples. Statistical analyses were performed using SPSS software (version 22.0; SPSS Inc., Chicago, IL, USA). A threshold of *p* < 0.05 was considered statistically significant.

## 3. Results

### 3.1. Weight and Blood Glucose Measurement

Six randomly selected rats were subjected to measurements of weight and blood glucose levels before fasting, after fasting, and at 1, 4, 7,10, and 14 days after STZ injection. Before induction of type 1 diabetes, all rats had blood glucose levels that remained between 100 and 130 mg/dL before fasting and decreased to <100 mg/dL after fasting. Blood glucose levels remained ≥350 mg/dL for 14 days after STZ injection. In Figure 5, there are no changes in body weight between measurements before fasting and measurements at 14 days after STZ injection.

### 3.2. Confirmation of Wound Size in Animal Diabetic Wound Model Exposed to PM 2.5

Over time, wound size decreased in all groups. On day 4 after exposure to 80 µg/m^3^ DEPs, wound size decreased in the following order: group 1 (73.10%) > normal control (77.12%) > group 2 (82.18%). On day 7, wound size decreased as follows: group 1 (53.87%) > normal control (56.75%) > group 2 (68.51%). On day 10, wound size decreased as follows: normal control (22.30%) > group 2 (26.15%) > group 1 (30.59%). On day 14, wound size decreased in the order of group 2 (14.31%) > group 1 (14.68%) > normal control (16.19%); however, statistical analysis showed no significant difference (*p* > 0.05). The data are shown in Figure 6 and Table 5.

### 3.3. Western Blot

Protein levels of the cytokines MMP-9, TIMP-1, TNF-α, and NF-κB p65 were measured in Figure 7 and Table 6. In Figure 8, the level of MMP-9 expression was low in group 1 beginning on day 7, whereas the normal control group and group 2 had higher levels of MMP-9 expression; however, these differences were not statistically significant (*p* > 0.05). In Figure 9 and Figure 10, group 2 showed lower levels of TIMP-1 and TNF-α expression compared with group 1, but these differences were not statistically significant (*p* > 0.05). In Figure 11, the levels of NF-κB p65 expression were similar in all groups; on day 14, group 1 tended to exhibit increased expression, although this difference was not statistically significant (*p* > 0.05).

### 3.4. Hematoxylin and Eosin (H&E) Staining

Wound tissue collected on days 4, 7, 10, and 14 was stained with H&E; granulation tissue formation and inflammatory cell infiltration are scored in Figure 12 and Table 7 and Table 8. In Figure 13, in the normal control group, granulation tissue formation was not observed on days 4–10 after injury, although it was observed on day 14. In group 1, granulation tissue formation appeared on day 4, but the degree of granulation tissue formation gradually decreased over time. In group 2, granulation tissue was not observed on days 4–10 as in the control group; however, granulation tissue was observed on day 14. In group 1, many inflammatory cells were observed beginning on day 7, whereas fewer inflammatory cells were observed in the normal control group and group 2.

### 3.5. Immunohistochemistry (IHC) Staining

Wound tissue collected on days 4, 7, 10, and 14 was subjected to IHC staining; the staining intensity and percentage of inflammatory cells were scored. In Figure 14, with respect to MMP-9, group 1 showed a tendency toward weaker expression than the normal control group. MMP-9 expression was higher in group 2 than in either of the other two groups. In Figure 15, TIMP-1 expression was similar in all groups; on day 14, many cells exhibited TIMP-1-positive staining in group 2. TNF-α expression was higher in group 1 than in the normal control group; notably, TNF-α expression was higher in group 2. In Figure 16, many cells exhibited TNF-α-positive staining in group 2. In Figure 17, NF-κB expression was lower in group 1 and the normal control group than in group 2 on day 4, but it was higher in group 2 on day 10. The scoring data are shown in Figure 18 and Table 9.

## 4. Discussion

In general, when wounds occur, macrophages recruit and activate white blood cells to secrete cytokines that promote inflammatory reactions [19]. Activated macrophages are classified into M1 and M2 subtypes. M1 macrophages produce pro-inflammatory cytokines such as IL-6, IL-12, TNF-α, MMP-2, and MMP-9, which promote the production of reactive oxygen species and nitric oxide to combat potential infection; in contrast, M2 macrophages can produce low levels of anti-inflammatory cytokines (IL-10, IL-14, and transforming growth factor-β) that promote the repair of damaged tissue [20,21,22]. The pro-inflammatory cytokine TNF-α plays an important role in wound healing by regulating cell death pathways(e.g., NF-κB activation) [23]. NF-κB is associated with skin wound healing, cell proliferation, adhesion, and the removal of reactive oxygen species. NF-κB signaling pathways and wound healing processes are closely associated at the molecular level [24]. M1 macrophages also secrete MMP-2 and MMP-9, which help to create a space for the decomposition of extracellular substrates and migration of inflammatory cells. M2 macrophages produce TIMP-1, which inhibits MMP activity and produces extracellular matrix [25]. During the progression of wound healing, M1 macrophages are converted into M2 macrophages; in chronic wounds (e.g., diabetic wounds), the M1 to M2 transition is not properly controlled, leading to delayed healing [26,27].

Recently, as problems caused by fine dust have increased, many problems have occurred, and diabetics are more adversely affected than ordinary patients. When diabetic rats are exposed to fine dust, oxidative stress occurs, increasing the expression of infectious cytokines such as IL-1b, TNF-a, and IL-6 and decreasing the expression of anti-inflammatory cytokines such as IL-10, cyclooxygenase-2 (COX-2), delaying wound healing, and inhibiting glucose absorption in the pancreas [28,29]. When diabetic rats are exposed to DEPs, oxidative stress occurs; the expression levels of pro-inflammatory cytokines such as TNF-α, IL-6, and cyclooxygenase-2 (COX-2) increase, delaying the healing of diabetic wounds [30,31]. Thus, there is a need for materials that can reduce infection in diabetic wounds exposed to DEPs and efficiently heal wounds.

KRG stimulates insulin release in a glucose-independent manner by inhibiting the K+ channel (KATP channel), depolarizing β-cells in the pancreas, and stimulating Ca2+ inflow, thereby improving blood sugar concentration and insulin sensitivity [32,33]. Ginsenoside Rb1, one of the components of red ginseng, stimulates the production of VEGF to promote angiogenesis, and promotes skin regeneration by increasing the expression of HIF-1α and IL-1β at the keratin site [34].

Probiotics reduce insulin resistance and oxidative stress by increasing SOD (superoxide dismutase), GSH (glutathione), and catalase (CAT) activity, and reduce the expression of inflammatory cytokine tissue necrosis factor (TNF)-α, interleukin (IL)-1β, and IL-6 [35,36]. In addition, probiotics promote the formation of parenting tissues, promote the proliferation and movement of keratinocytes, and promote wound healing by activating the expression of IL-8 [37,38].

Therefore, this study confirmed that a mixture of Korean red ginseng and probiotics, which are known to be effective in treating diabetes and general wounds, can help heal wounds by controlling the expression of TNF-α, MMP-9, TIMP-1, and NF-κB.

The expression of TNF-α is upregulated in diabetic rats; the long-term upregulation of TNF-α can delay wound healing by increasing the production of MMP-9 and inhibiting the synthesis of extracellular matrix proteins and TIMP-1 [39,40].

Rats treated with diabetes and DEPs reported increased TNF-a expression compared to rats not treated with DEPs [41], and the results of this study also showed elevated TNF-α expression in group 1, a group treated with DEPs, while rats treated with probiotics and KRG had reduced TNF-α expression.

A balanced expression of MMP and TIMP proteins promotes wound healing, whereas an elevated expression of TIMP or MMP proteins delays wound healing [42,43]. This study confirmed that TIMP-1 expression was elevated in rats that had been exposed to 80 μg/m^3^ DEPs; it decreased to baseline in rats that had received probiotic and KRG treatment. In the case of diabetic wounds, the damage to keratinocytes continues the inflammatory process characterized by oxidative stress that can cause dysfunction and cell death [44]. Histological analysis of this study showed no granulocyte formation after exposure to 80 μg/m^3^ DEPs.

Histology analysis confirmed that granulocyte formation was not observed after exposure to 80 μg/m^3^ DEPs; wound healing was delayed because of the increased presence of inflammatory cells. In contrast, rats administered with probiotics and KRG showed granulocytosis on the 14th day, and the number of inflammatory cells decreased compared to rats in the normal control group.

None of the results of this study showed significant differences, but exposure to DEPs increased the expression of anti-inflammatory cytokines such as TNF-a and NF-κB and decreased the expression of anti-inflammatory cytokines, while the group applied a combination of probiotics and Korean red ginseng tended to decrease the expression of anti-inflammatory cytokines. These results suggest that applying a combination of probiotics and Korean red ginseng to diabetic wounds exposed to fine dust can accelerate wound healing.

This study may not reflect the complexity of diseases clinically observed in animal models and experiments. Since the number of samples used in this study was small, additional experiments are needed to more accurately confirm the effect of the combination of probiotics and red ginseng.

## 5. Conclusions

Combined treatment involving probiotics and KRG promotes skin regeneration by controlling granule tissue formation and cytokine expression during wound healing. Therefore, a combination of probiotics and KRG may be an effective treatment for DEP-exposed diabetic wounds.

## Figures and Tables

**Figure 1 medicina-59-01155-f001:**
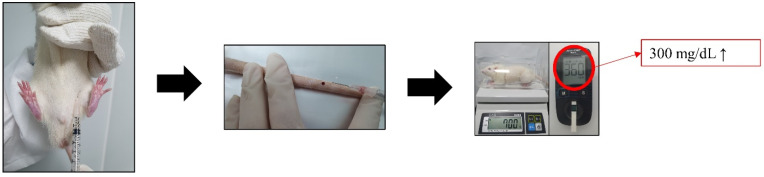
Induction of type 1 diabetes by injection of STZ.

**Figure 2 medicina-59-01155-f002:**
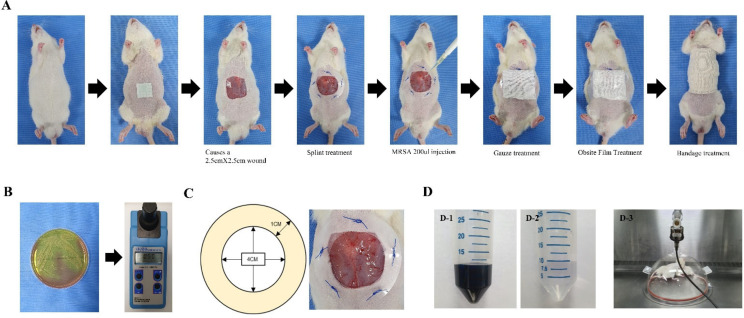
Protocol for generation of MRSA-infected diabetic wounds exposed to PM 2.5: (**A**) The process of creating a diabetic wound animal model; (**B**) MRSA culture; (**C**) splint; (**D**) DEP inhalation; (**D-1**) DEP stock solution; (**D-2**) DEP 80 μg/m^3^; (**D-3**) DEP inhalation.

**Figure 3 medicina-59-01155-f003:**
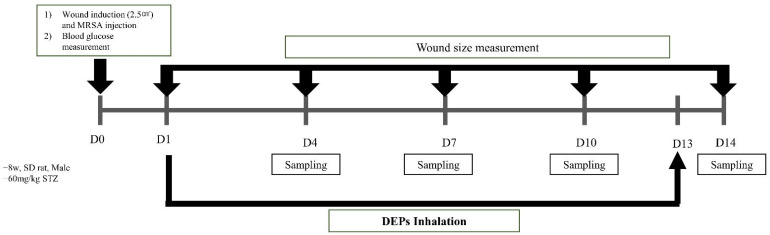
Protocol for inhalation of diesel exhaust particles (DEPs).

**Figure 4 medicina-59-01155-f004:**
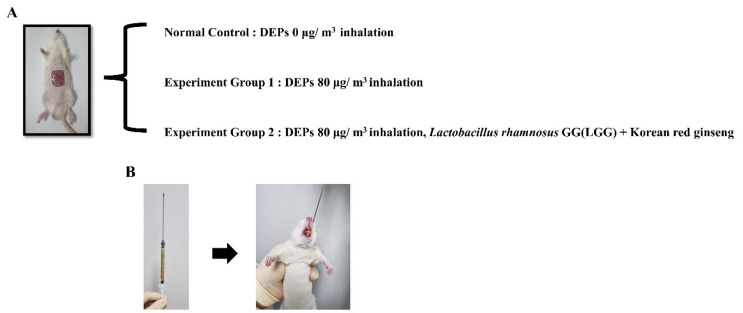
Study design and combination treatment involving probiotics and Korean red ginseng for DEP-exposed diabetic wounds: (**A**) Study design. (**B**) Probiotics + Korean red ginseng treatment.

**Figure 5 medicina-59-01155-f005:**
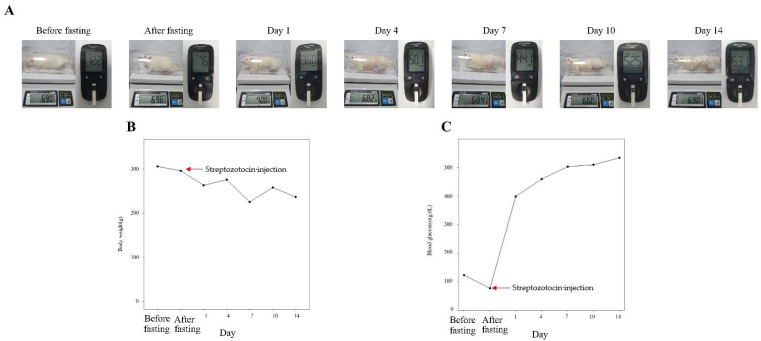
Body weight and blood glucose in STZ-induced diabetic rats. Body weight and blood glucose were measured for 14 days after induction of type 1 diabetes by STZ injection (60 mg/kg). (**A**) Body weight and blood glucose. (**B**) Body weight. (**C**) Blood glucose. Number of animals = 6.

**Figure 6 medicina-59-01155-f006:**
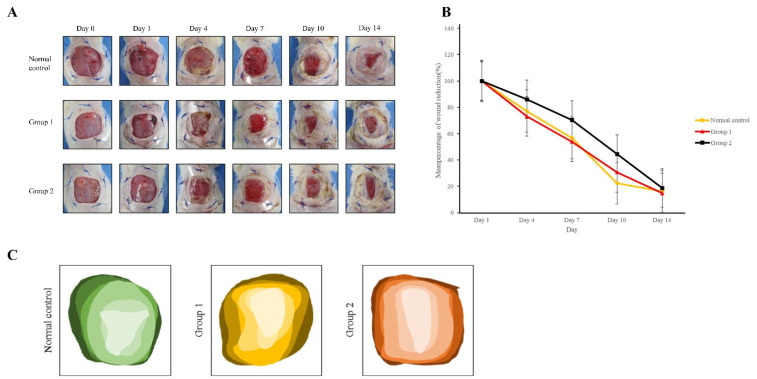
Wound size measurement: (**A**) Representative images. (**B**) Wound size reduction rates (*n* = 3). (**C**) Progression of wound closure.

**Figure 7 medicina-59-01155-f007:**
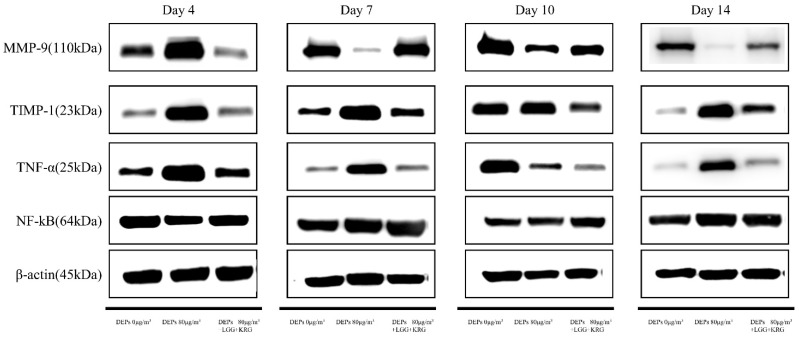
Expression levels of macrophage-related cytokine genes in DEP-exposed diabetic wounds treated with probiotics and Korean red ginseng.

**Figure 8 medicina-59-01155-f008:**
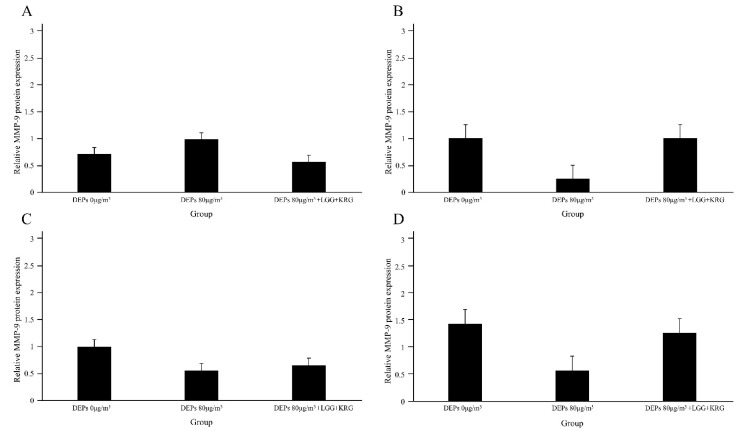
Expression levels of MMP-9 in DEP-exposed diabetic wounds: (**A**) Day 4. (**B**) Day 7. (**C**) Day 10. (**D**) Day 14.

**Figure 9 medicina-59-01155-f009:**
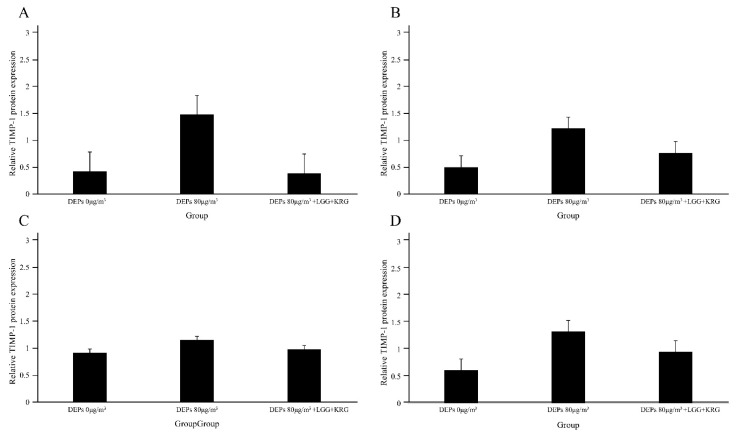
Expression levels of TIMP-1 in DEP-exposed diabetic wounds: (**A**) Day 4. (**B**) Day 7. (**C**) Day 10. (**D**) Day 14.

**Figure 10 medicina-59-01155-f010:**
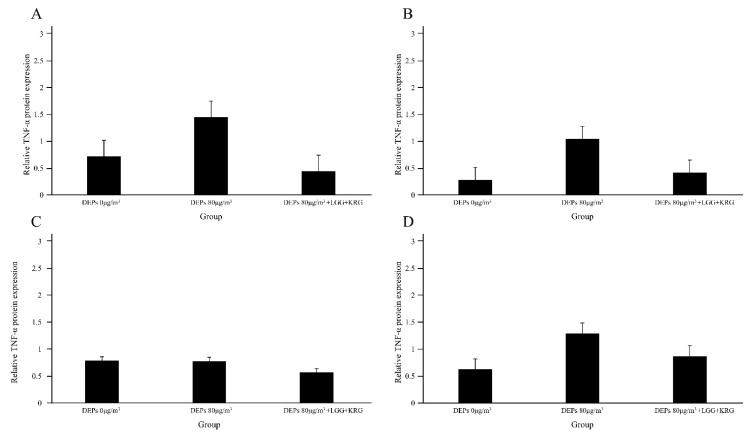
Expression levels of TNF-α in DEP-exposed diabetic wounds: (**A**) Day 4 (**B**) Day 7 (**C**) Day 10 (**D**) Day 14.

**Figure 11 medicina-59-01155-f011:**
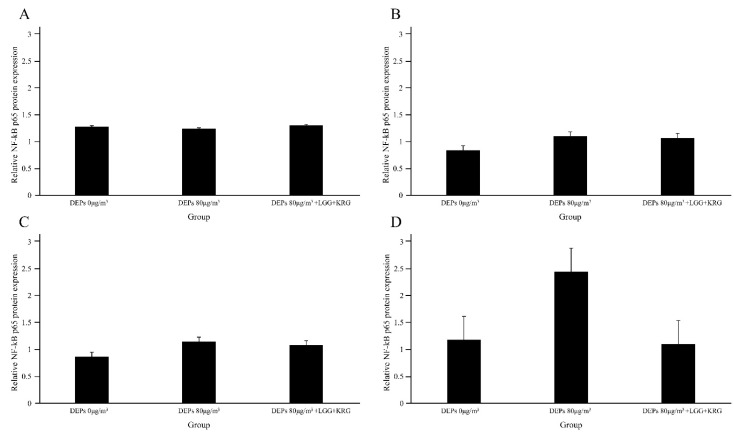
Expression levels of NF-κB p65 in DEP-exposed diabetic wounds: (**A**) Day 4. (**B**) Day 7. (**C**) Day 10. (**D**) Day 14.

**Figure 12 medicina-59-01155-f012:**
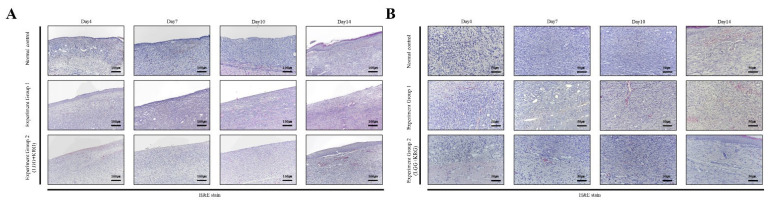
Histology staining analysis on days 4, 7, 10, and 14 after wound formation: (**A**) Magnification, X100. (**B**) Magnification, ×200.

**Figure 13 medicina-59-01155-f013:**
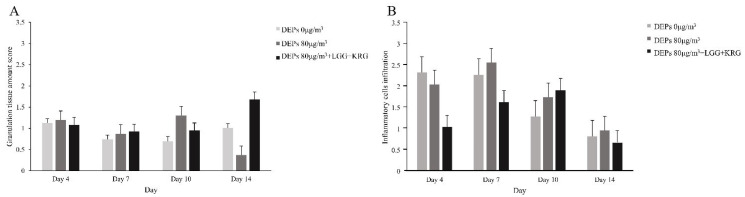
Histology staining analysis on days 4, 7, 10, and 14 after wounding: (**A**) Granulation tissue amount score. (**B**) Inflammatory cells infiltration score.

**Figure 14 medicina-59-01155-f014:**
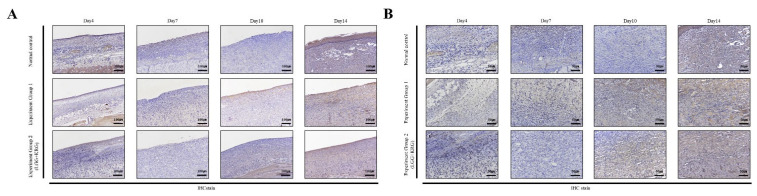
MMP-9 expression in DEP-exposed diabetic wounds, as determined by IHC staining: (**A**) Magnification, ×100. (**B**) Magnification, ×200.

**Figure 15 medicina-59-01155-f015:**
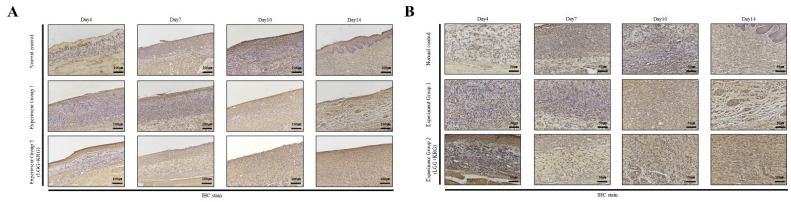
TIMP-1 expression in DEP-exposed diabetic wounds, as determined by IHC staining: (**A**) Magnification, ×100. (**B**) Magnification, ×200.

**Figure 16 medicina-59-01155-f016:**
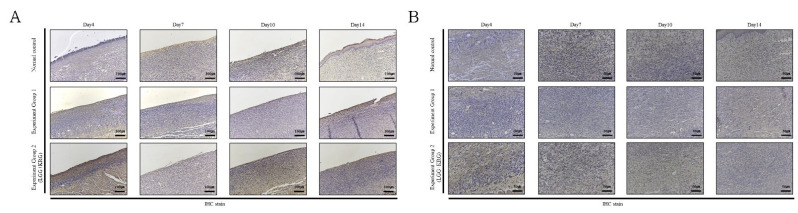
TNF-α expression in DEP-exposed diabetic wounds, as determined by IHC staining: (**A**) Magnification, ×100. (**B**) Magnification, ×200.

**Figure 17 medicina-59-01155-f017:**
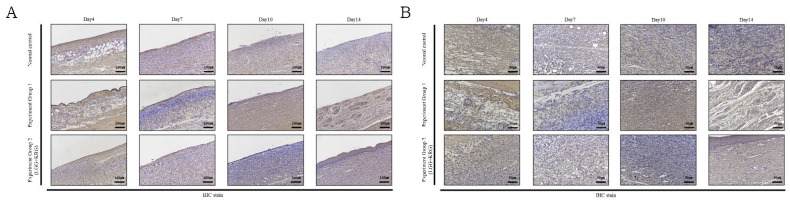
NF-κB expression in DEP-exposed diabetic wounds, as determined by IHC staining: (**A**) Magnification, ×100. (**B**) Magnification, ×200.

**Figure 18 medicina-59-01155-f018:**
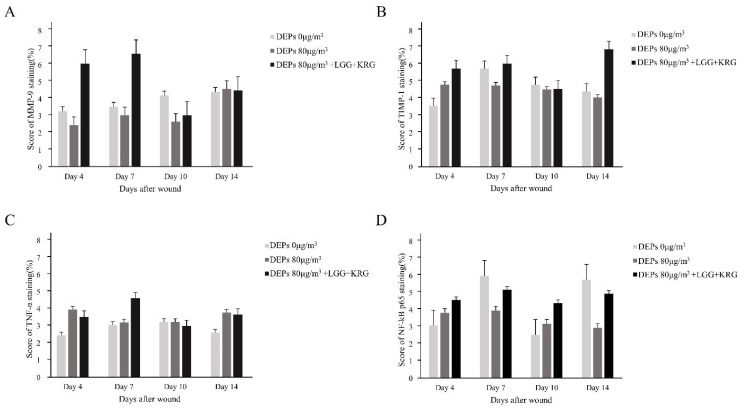
Cytokine expression in DEP-exposed diabetic wounds, as determined by IHC staining: (**A**) MMP-9. (**B**) TIMP-1. (**C**) TNF-α. (**D**) NF-κB.

**Table 1 medicina-59-01155-t001:** Western blot antibody.

Target Gene	Company	Dilution	Size
Anti NF-κB p65	Abcam	1:300	64 kDa
Anti TNF-A	Santa Cruz Biotechnology	1:500	25 kDa
Anti MMP-9	Abcam	1:500	92 kDa
Anti TIMP-1	Santa Cruz Biotechnology	1:1000	23 kDa
Anti B-actin	Cell Signaling	1:5000	45 kDa

**Table 2 medicina-59-01155-t002:** Granulation tissue formation score as determined by H&E staining.

Judgement Element	Standard	Score
1) Granulation tissue of formation	None (not formation)	0
2) Epidermis formation	Scant	1
3) Keratin layer formation	Moderate	2
4) Collagen deposition	Abundant	3

0 indicates low granulation and 3 indicates high granulation.

**Table 3 medicina-59-01155-t003:** Inflammatory cells infiltration as determined by H&E staining.

Judgement Element	Standard	Score
Inflammatory cells infiltration	None (no necrosis and inflammation)	0
	Slight inflammation (not clusters and only exist in cell form)	1
	Moderate inflammation (when small clusters are formed and inflammatory cells are visible)	2
	Abundant (when large clusters are formed and many inflammatory cells are visible)	3

0 indicates low and 3 indicates high inflammatory cell infiltration.

**Table 4 medicina-59-01155-t004:** Expression score of each cytokine by immunohistochemistry (IHC) staining.

Judgement Element	Standard	Score
Dyeing intensity	None	0
	Slight	1
	Moderate	2
	Heavy	3
Cell staining rate (%)	0%	0
	1–25%	1
	26–50%	2
	51–75%	3
	76–100%	4

The percentage of stained cells was scored from 0 to 4, with 0 indicating no staining and 4 a large proportion of stained cells. The two scores were then multiplied (range: 0–12 points).

**Table 5 medicina-59-01155-t005:** Wound closure rate(%).

Experimental Groups	Wound Closure Rate (%)					
	Period of Treatment					
	Day 1	Day 4	Day 7	Day 10	Day 14	N
Normal control (0 µg/m^3^)	100	77.12 ± 1.27	56.75 ± 16.61	22.31 ± 7.64	16.20 ± 7.60	3
Group 1 (80 µg/m^3^)	100	61.55 ± 31.69	32.17 ± 21.94	21.06 ± 15.16	11.82 ± 5.15	3
Group 2 (80 µg/m^3^,LGG + KRG)	100	86.06 ± 8.97	70.34 ± 5.09	44.47 ± 1.52	18.82 ± 3.35	3

Data are mean ± SD for three rats in each group. Significance was determined using the Mann–Whitney U test.

**Table 6 medicina-59-01155-t006:** Expression levels of NF-κB p65 in DEP-exposed diabetic wounds.

Experimental Groups	Relative Protein Expression (%)					
	Period of Treatment					
	Gene	Day 4	Day 7	Day 10	Day 14	N
Normal control (0 µg/m^3^)	MMP-9	0.71 ± 0.59	1.01 ± 0.31	1.00 ± 0.18	1.45 ± 0.13	3
	TIMP-1	0.42 ± 0.24	0.50 ± 0.03	0.92 ± 0.34	0.62 ± 0.46	3
	TNF-a	0.72 ± 0.35	0.2 ± 0.06	0.79 ± 0.41	0.63 ± 0.46	3
	NF-kB	1.28 ± 0.20	0.84 ± 0.37	0.87 ± 0.28	1.18 ± 0.40	3
Group 1 (80 µg/m^3^)	MMP-9	0.99 ± 0.23	0.25 ± 0.26	0.56 ± 0.51	0.58 ± 0.46	3
	TIMP-1	1.48 ± 0.82	1.22 ± 0.04	1.15 ± 0.20	1.33 ± 0.26	3
	TNF-a	1.45 ± 0.96	1.05 ± 0.17	0.78 ± 0.33	1.29 ± 0.40	3
	NF-kB	1.24 ± 0.32	1.10 ± 0.49	1.15 ± 0.59	2.45 ± 1.00	3
Group 2 (80 µg/m^3^,LGG + KRG)	MMP-9	0.57 ± 0.27	1.01 ± 0.23	0.65 ± 0.26	1.37 ± 0.24	3
	TIMP-1	0.39 ± 0.27	0.77 ± 0.27	0.98 ± 0.24	0.96 ± 0.43	3
	TNF-a	0.44 ± 0.19	0.42 ± 0.23	0.57 ± 0.31	0.87 ± 0.66	3
	NF-kB	1.30 ± 0.24	1.07 ± 0.20	1.08 ± 0.31	1.10 ± 0.01	3

Data are mean ± SD for three rats in each group. Significance was determined using the Mann–Whitney U test.

**Table 7 medicina-59-01155-t007:** Granulation tissue of formation score by hematoxylin and eosin (H&E) staining.

Experimental Groups	Granulation Tissue Formation (%)				
	Period of Treatment				
	Day 4	Day 7	Day 10	Day 14	N
Normal control (0 µg/m^3^)	1.38 ± 0.38	0.74 ± 0.42	0.70 ± 0.26	1.01 ± 0.26	3
Group 1 (80 µg/m^3^)	1.20 ± 0.83	0.88 ± 0.45	1.31 ± 0.45	0.38 ± 0.23	3
Group 2 (80 µg/m^3^,LGG + KRG)	0.95 ± 0.48	0.93 ± 0.42	0.95 ± 0.63	1.68 ± 0.62	3

Data are mean ± SD for three rats in each group. Significance was determined using the Mann–Whitney U test.

**Table 8 medicina-59-01155-t008:** Inflammatory cells infiltration score by hematoxylin and eosin (H&E) staining.

Experimental Groups	Inflammatory Cells Infiltration (%)				
	Period of Treatment				
	Day 4	Day 7	Day 10	Day 14	N
Normal control (0 µg/m^3^)	1.27 ± 0.19	2.41 ± 0.15	1.61 ± 0.25	1.10 ± 0.18	3
Group 1 (80 µg/m^3^)	2.09 ± 0.38	2.63 ± 0.14	1.83 ± 0.36	1.19 ± 0.09	3
Group 2 (80 µg/m^3^, LGG + KRG)	1.16 ± 0.14	1.74 ± 0.47	1.91 ± 0.46	0.98 ± 0.34	3

Data are mean ± SD for three rats in each group. Significance was determined using the Mann–Whitney U test.

**Table 9 medicina-59-01155-t009:** Inflammatory cells infiltration score by hematoxylin and eosin (H&E) staining.

Experimental Groups	Score of IHC Staining(%)					
	Period of Treatment					
	Gene	Day 4	Day 7	Day 10	Day 14	N
Normal control (0 µg/m^3^)	MMP-9	3.29 ± 1.88	3.56 ± 0.51	4.21 ± 1.70	4.42 ± 1.86	3
	TIMP-1	3.60 ± 1.41+	5.78 ± 1.06	4.83 ± 1.82	4.44 ± 0.98	3
	TNF-a	2.46 ± 0.64	3.07 ± 0.29	3.24 ± 0.91	2.62 ± 0.91	3
	NF-kB	3.08 ± 0.52	6.00 ± 0.00	2.54 ± 0.36	5.77 ± 2.35	3
Group 1 (80 µg/m^3^)	MMP-9	2.48 ± 0.58	3.05 ± 0.20	2.67 ± 0.47	4.59 ± 3.19	3
	TIMP-1	4.83 ± 2.89	4.79 ± 1.39	4.55 ± 1.26	4.08 ± 1.21	3
	TNF-a	3.96 ± 0.73	3.21 ± 0.59	3.23 ± 0.91	3.79 ± 1.61	3
	NF-kB	3.83 ± 0.38	3.96 ± 0.94	3.19 ± 0.05	2.96 ± 0.44	3
Group 2 (80 µg/m^3^,LGG + KRG)	MMP-9	6.08 ± 3.17	6.66 ± 2.23	3.04 ± 1.06	4.50 ± 1.44	3
	TIMP-1	5.79 ± 1.58	6.08 ± 1.61	4.59 ± 0.95	6.92 ± 2.06	3
	TNF-a	3.54 ± 0.92	4.63 ± 1.20	3.00 ± 0.25	3.67 ± 0.88	3
	NF-kB	4.58 ± 0.38	5.19 ± 2.19	4.41 ± 0.96	4.95 ± 0.88	3

Scoring of cytokine expression by immunohistochemistry (IHC) staining.

## Data Availability

Not applicable.

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
