# Peer review of "Effect of the Combination of Probiotics and Korean Red Ginseng on Diabetic Wound Healing Exposed to Diesel Exhaust Particles(DEPs)"

_medicina, 2023, doi:10.3390/medicina59061155_

Round 1
Reviewer 1 Report
I reviewed the manuscript entitled Effect of combination of probiotics and Korean red ginseng on diabetic wound healing exposed to diesel exhaust particles(DEPs).
The author reported interesting experimental results: a combination of probiotics and KRG may be an effective treatment for DEPs exposed diabetic wounds.
I agree to accept this manuscript, but there are a few issues that need to be revised before acceptance.
1) Figure 1. 300mg/dl should change to 300 mg/dL. There should be spaces between numbers and units.
2) Figure 2 Line 119, (D-3) DEPs inlataion, is there a spelling error here? Inhalation is right.
3) Figure 4 is not seen in the text.
4) Table 1, NF-kB p65, k is incorrect, κ is right.
5) 2.1. Animal and induction of diabetes, the author mentioned using rats, but in the discussion section, they also mentioned using mice in Line 342, the author should review the entire text and correct any errors.
6) In the references, many references only lists one author. Some have listed multiple authors, please unify the citation format of the references.
7) Such as ref 27, The starting and ending page numbers of the references are incomplete and should be listed in full. Such as ref 31, 201-210 is right.
Minor editing of English language required
Author Response
Thank you very much for reviewing our paper. I sincerely appreciate and totally agree with all your comments. Using them, I have done my best to correct the manuscript.
1) Figure 1. 300mg/dl should change to 300 mg/dL. There should be spaces between numbers and units.
->We appreciate for your comments. We have corrected that Figure..
2) Figure 2 Line 119, (D-3) DEPs inlataion, is there a spelling error here? Inhalation is right.
->We appreciate for your comments. We have corrected that word correctly.
3) Figure 4 is not seen in the text.
->We appreciate for your comments. I attached the Figure 4 again.
4) Table 1, NF-kB p65, k is incorrect, κ is right.
->We appreciate for your comments. We have corrected that word correctly.
5) 2.1. Animal and induction of diabetes, the author mentioned using rats, but in the discussion section, they also mentioned using mice in Line 342, the author should review the entire text and correct any errors.
->We appreciate for your comments. We have corrected that word correctly.
6) In the references, many references only lists one author. Some have listed multiple authors, please unify the citation format of the references.
->Thank you for your opinion. After checking the manuscript, the part marked with "mice" was revised to "rat".
7) Such as ref 27, The starting and ending page numbers of the references are incomplete and should be listed in full. Such as ref 31, 201-210 is right.
->I agree with you. However, as a result of checking the paper, it started at 419pages and ended at 419 pages, and accordingly, it was marked as 419 p(ref 26).

Reviewer 2 Report
Review
In this work, the effect of combination of probiotics and korean red ginseng on diabetic wound model exposed 16 to DEPs was investigated. Rats were used for animal model experiments. The results showed that combined treatment with PB and KRG can promote the healing of DEP-exposed diabetic wounds. This paper is thorough, well-structured and likely of interest to readers focused on diabetic wound healing. Before publication, it is necessary to clarify following points.
Questions
1. In Figure 6, the wound closure progress should be further displayed for readers. The author could finish it by using Image J software. Please read and cite the following reference (Figure 5B).
[1]Shixiong Yi, Ying Zhou, Jiamei Zhang, et al. Natural Flat Silk Cocoon-Based Dressing: Daylight-Driven and Rechargeable Antibacterial Membranes Accelerate the Infected Wound Healing. Advanced Healthcare Materials, 2022: 2201397.
2. The author should consider whether it is necessary to supplement the cytotoxicity test (MTT).
3. The author should carefully check this manuscript. For example, in part 2.5. Wound size measurement, ‘A square shape with a size of 1 cm2’. The ‘cm2’ should be revised to be ‘cm2’.
Minor editing of English language required
Author Response
Thank you very much for reviewing our paper. I sincerely appreciate and totally agree with all your comments. Using them, I have done my best to correct the manuscript.
- In Figure 6, the wound closure progress should be further displayed for readers. The author could finish it by using Image J software. Please read and cite the following reference (Figure 5B).
[1]Shixiong Yi, Ying Zhou, Jiamei Zhang, et al. Natural Flat Silk Cocoon-Based Dressing: Daylight-Driven and Rechargeable Antibacterial Membranes Accelerate the Infected Wound Healing. Advanced Healthcare Materials, 2022: 2201397.
->We appreciate for your comments. A clear picture of wound closure progress will help the reader understand this study. After modifying Figure 6, we added an explanation of what the following figure measures.
- The author should consider whether it is necessary to supplement the cytotoxicity test (MTT).
->Thank you for your opinion. However, since this study aims to confirm the efficacy of red ginseng and probiotics in vivo, it is unlikely that a cytotoxicity test will be required.
- The author should carefully check this manuscript. For example, in part 2.5. Wound size measurement, ‘A square shape with a size of 1 cm2’. The ‘cm2’ should be revised to be ‘cm2’.
->Thank you for your opinion and sorry to repeat simple mistakes. We checked the manuscript again and corrected it correctly.

Reviewer 3 Report
Dear author/ editor,
I read carefully the MS Effect of Combination of Probiotics and Korean Red Ginseng on Diabetic Wound Healing Exposed to Diesel Exhaust Particles(DEPs), where the effect of combination of probiotics and korean red ginseng was investigated on diabetic wound model exposed to DEPs.
According to my opinion, there are major issues that are not clear in the MS:
1. The hypothesis and the aim of the study is poor or should be improved.
2. The wound are mechanically induced, not by diabetes by itself, so, why it is necessary to induce diabetes?
3. The connection between diabetes and DEPs and probiotics is not clear and is insufficient in literature.
4. There only one WB measurement, that’s why there is no Standard deviation on the results.
5. Most of the results are not significant.
6. Some part of the figures are not necessary and sufficient.
Best regards
Author Response
Thank you very much for reviewing our paper. I sincerely appreciate and totally agree with all your comments. Using them, I have done my best to correct the manuscript.
- The hypothesis and the aim of the study is poor or should be improved.
->We agree with your comments. Based on your comments, the hypothesis and purpose of the study were added to the introduction.
- The wound are mechanically induced, not by diabetes by itself, so, why it is necessary to induce diabetes?
->Thank you for your comment. Since diabetic complications cause foot ulcers, a diabetic infectious disease, we caused diabetes to find a treatment for diabetic infection injuries. Also, I added this part to the introduction part.
- The connection between diabetes and DEPs and probiotics is not clear and is insufficient in literature.
->Thank you for your comment. I added this part to the discussion part.
- There only one WB measurement, that’s why there is no Standard deviation on the results.
->Thank you for your comment. WB measurement was conducted with three samples, so there is a standard deviation. Regarding this part, I added it to table.
- Most of the results are not significant.
->Thank you for your comment. I added this information to the discussion section.
- Some part of the figures are not necessary and sufficient.
->Thank you for your comment. Added numerical values for experimental data.

Round 2
Reviewer 3 Report
Dear Author/ Editor,
I accept the publication of the MS in the corrected version.